

# Genome-wide identification and characterization of TCP family genes in *Brassica juncea* var. tumida

Jing He[1,*], Xiaohong He[1,*], Pingan Chang[1], Huaizhong Jiang[1], Daping Gong[2] and Quan Sun[1]

[1] Chongqing University of Posts and Telecommunications, College of Bioinformation, Chongqing Key Laboratory of Big Data for Bio Intelligence, ChongQing, China
[2] Tobacco Research Institute of Chinese Academy of Agricultural Sciences, Qingdao, China
[*] These authors contributed equally to this work.

## ABSTRACT

**Background**. Teosinte branched1/*Cycloidea*/proliferating cell factors (TCPs) are plant-specific transcription factors widely involved in leaf development, flowering, shoot branching, the circadian rhythm, hormone signaling, and stress responses. However, the TCP function in *Brassica juncea* var. tumida, the tumorous stem mustard, has not yet been reported. This study identified and characterized the entire TCP family members in *B. juncea* var. tumida.

**Methods**. We identified 62 BjTCP genes from the *B. juncea* var. tumida genome and analyzed their phylogenetic relationship, gene structure, protein motifs, chromosome location, and expression profile in different tissues.

**Results**. Of the 62 *BjTCP* genes we identified in *B. juncea* var. tumida, containing 34 class I and 28 class II subfamily members, 61 were distributed on 18 chromosomes. Gene structure and conserved motif analysis showed that the same clade genes displayed a similar exon/intron gene structure and conserved motifs. Cis-acting element results showed that the same clade genes also had a similar cis-acting element; however, subtle differences implied a different regulatory pathway. The BjTCP18s members were low-expressed in Dayejie strains and the unswelling stage of Yonganxiaoye strains. Treatment with gibberellin (GA) and salicylic acid (SA) showed that GA and SA affect the expression levels of multiple *TCP* genes.

**Conclusion**. We performed the first genome-wide analysis of the TCP gene family of *B. juncea* var. tumida. Our results have provided valuable information for understanding the classification and functions of *TCP* genes in *B. juncea* var. tumida.

# INTRODUCTION

The teosinte branched1/*Cycloidea*/proliferating cell factor (TCP) family is a group of plant-specific transcription factors (TFs) reportedly involved in embryonic growth (*Takeda et al., 2003*), leaf development (*Bresso et al., 2018*; *Danisman et al., 2012*; *Du et al., 2017*; *Kieffer et al., 2011*; *Liu et al., 2018b*; *Ma et al., 2016*; *Uberti-Manassero et al., 2012*; *Wang et al.,*

Corresponding authors
Daping Gong, gongdaping@caas.cn
Quan Sun, sunquan@cqupt.edu.cn

*2018*), branching (*Aguilar-Martinez, Poza-Carrion & Cubas, 2007*; *Brewer, 2015*; *Dixon et al., 2018*; *Gonzalez-Grandio et al., 2013*; *Gonzalez-Grandio et al., 2017*; *Martin-Trillo et al., 2011*; *Maurya et al., 2020*; *Niwa et al., 2013*; *Seale, Bennett & Leyser, 2017*; *Shen et al., 2019*; *Wang et al., 2019a*), flowering (*Aguilar-Martinez, Poza-Carrion & Cubas, 2007*; *Damerval et al., 2007*; *Finlayson, 2007*; *Madrigal, Alzate & Pabon-Mora, 2017*; *Navarro, Cruz-Oro & Prat, 2015*; *Yang et al., 2015*), the circadian rhythm (*Beveridge et al., 2003*; *Giraud et al., 2010*), hormone signaling (*Wang et al., 2019a*), and stress responses (*Danisman, 2016*; *Guan et al., 2017*; *Martin-Trillo & Cubas, 2010*). The TCP domain is highly conserved in the TCP family and is formed by an N-terminal region enriched in basic amino acids, followed by two amphipathic α-helices connected by a disordered loop (*Cubas, Vincent & Coen, 1999*; *Doebley, Stec & Hubbard, 1997*).

On the basis of this conserved domain, TCP proteins are divided into two subfamilies, class I and class II (*Li, 2015*; *Martin-Trillo & Cubas, 2010*). The difference between the two classes is the deletion of four amino acids in the TCP domain in class I. In *Arabidopsis*, TCP2–5, TCP10, TCP13, TCP17, and TCP24 are related to lateral organ organogenesis and control leaf development (*Efroni et al., 2008*; *Hay, Barkoulas & Tsiantis, 2004*; *Koyama et al., 2007*; *Qin et al., 2005*). Belonging to the same subfamily, branched1 (TCP18) and branched2 (TCP12) play an important role in controlling branch outgrowth (*Aguilar-Martinez, Poza-Carrion & Cubas, 2007*; *Gonzalez-Grandio et al., 2013*; *Muhr et al., 2016*; *Wang et al., 2019a*). By interacting with florigen proteins FLOWRING LOCUS T (FT), TCP18 inhibits floral transition of axillary meristems in *Arabidopsis* (*Niwa et al., 2013*). The *BRC1* homolog gene in the hybrid aspen also mediates photoperiodic control of seasonal growth (*Maurya et al., 2020*), and *TIG1*, encoding a TCP TF, contributes to plant architecture domestication in rice (*Zhang et al., 2019*). TCP21 participates in the circadian rhythm by binding to TIMING OF CAB EXPRESSION 1 (TOC1) and the CIRCADIAN AND CLOCK ASSOCIATED1 (CCA1) promoter (*Pruneda-Paz et al., 2009*). In addition, TCP proteins, such as brassinosteroids (BRs), jasmonic acid, indole-3-acetic acid (IAA), and strigolactone (SL), involved in plant growth and development, are usually regulated by phytohormone synthesis and metabolism (*Braun et al., 2012*; *Danisman et al., 2012*; *Li, 2015*; *Liu et al., 2017*; *Muhr et al., 2016*; *Qin et al., 2005*; *Schommer et al., 2008*). Studies have also reported on *TCP* genes regulated by sugars (*Wang et al., 2019b*) and light (*Kebrom, Burson & Finlayson, 2006*).

Recently, TCP proteins have been shown to be related to defense responses. For example, TCP13, TCP14, and TCP19 are directly targeted by effectors from *Pseudomonas syringae* and *Hyaloperonospora arabidopsidis* (*Mukhtar et al., 2011*). *Kim et al. (2014)* reported that TCP8, TCP13, TCP15, TCP20, TCP22, and TCP23 can interact with the *Arabidopsis* immune adaptor SUPPRESSOR OF rps4-RLD1 (SRFR1), which is a negative regulator of effector-triggered immunity (*Kim et al., 2014*). *TCP* genes are also regulated by microRNA 319 (miR319) and are involved in leaf development in *Arabidopsis* (*Bresso et al., 2018*; *Palatnik et al., 2003*; *Schommer et al., 2008*; *Wang et al., 2018*).

The TCP family has been identified in many different plant species, such as 24 *TCP* genes in *Arabidopsis* (*Martin-Trillo & Cubas, 2010*), 28 in *Oryza sativa*, 30 in *Lycopersicon esculentum* (*Parapunova et al., 2014*), 33 in *Populus euphratica* (*Ma et al., 2016*), 27 in

*Citrullus lanatus* s (*Shi et al., 2016*), 66 in *Triticum aestivum* (*Zhao et al., 2018*), 75 in *Gossypium barbadense* (*Zheng et al., 2018*), 31 in *Solanum tuberosum* (*Wang et al., 2019c*), 39 in *Brassica rapa L. ssp. Pekinensis* (*Liu et al., 2018b*), and 39 in *B. rapa ssp. rapa* (*Du et al., 2017*). *Liu et al. (2019)* performed a genome-wide systematic identification of the TCP proteins in the major plant lineages (47 species).

The tumorous stem mustard (*B. juncea* var. tumida) is an important crop of great economic value in China, so improving its yield is key issue for the Chinese pickle industry. The growth of *B. juncea* var. tumida involves four stages: germination, seedling, stem swelling, and flowering. Stem swelling is essential for tumorous stem formation, and the stem swelling–flowering balance is directly related to the quality and yield of tumorous mustards. *B. juncea* var. tumida is an annual plant, and for stem swelling, it is essential that the seeds be sown between mid-September and mid-October in Chongqing and other valleys of the Yangtze River, China. Therefore, the production period of edible stems is limited.

TCP proteins are extensively involved in branching, flowering, development, and plant morphology (*Aguilar-Martinez, Poza-Carrion & Cubas, 2007*; *Bai et al., 2012*; *Braun et al., 2012*; *Danisman et al., 2012*; *Dixon et al., 2018*; *Feng et al., 2018*; *Finlayson et al., 2010*; *Gonzalez-Grandio et al., 2013*; *Gonzalez-Grandio et al., 2017*; *Ho & Weigel, 2014*; *Li, 2015*; *Martin-Trillo et al., 2011*; *Nicolas et al., 2015*; *Niwa et al., 2013*; *Prusinkiewicz et al., 2009*; *Rameau et al., 2014*; *Seale, Bennett & Leyser, 2017*; *Teichmann & Muhr, 2015*). However, there are few reports on the TCP family in *B. juncea* var. tumida, and whether TCP proteins control stem swelling and flowering in *B. juncea* var. tumida is still unknown.

Since the entire genome of *B. juncea* var. tumida was sequenced (*Yang et al., 2016*), this study performed a genome-wide analysis of *TCP* genes for the first time. Of the 62 *BjTCP* genes identified, we analyzed their phylogenetic relationship, gene structure, protein motifs, chromosome location, and expression profile in different tissues. The results can provide valuable information for the classification of *BjTCP* genes and lay the foundation for exploring the molecular mechanism underlying stem swelling and flowering orchestrated by *TCP* genes in *B. juncea* var. tumida.

## MATERIAL AND METHODS

### Plant materials, growth conditions, and treatment

*B. juncea* var. tumida cultivar YA (with swollen tumorous stems) was used to analyze gene expression patterns. Seeds were sowed into 2:1 vermiculite:turfy soil and cultured at a constant temperature of 22 °C in a 16/8 h light/dark cycle in a culture room. Next, 3-week-old seedlings were used for exogenous hormone treatment; the seedlings were sprayed with 100 µM salicylic acid (SA) (*Feng et al., 2018*) and 100 µM gibberellin (GA) (*Rosa et al., 2017*). The second true leaf on each seedling was sampled at 0 (control), 2, 4, 6, 8, and 24 h after spraying. All treatments were repeated thrice, and each treatment was given to at least 20 seedlings. All materials were frozen immediately in liquid nitrogen and stored at −70 °C until RNA isolation.

## Identification of TCP proteins in *B. juncea* var. tumida

The genome sequences of *B. juncea* var. tumida (version 1.5), *B. nigra* (version 1.1), and *B. rapa* (version 3.0) were downloaded from the Brassica database (BRAD; http://brassicadb.org/brad/datasets/pub/Genomes/) (*Cheng et al., 2011*; *Yang et al., 2016*). In addition, the TCP domain in the Pfam database (accession no. PF03634) was downloaded (*Finn et al., 2010*), and the domain was searched in the BRAD using HMMER 3.0 with an $E$-value of $<1e^{-6}$ (*Finn, Clements & Eddy, 2011*). To confirm the results obtained by the HMMER algorithm, the TCP domain was further verified with Pfam and Smart databases (*Finn et al., 2010*; *Letunic & Bork, 2018*; *Letunic, Doerks & Bork, 2015*). The TCP protein sequences of *A. thaliana* were downloaded from the *Arabidopsis* information resource website (https://www.arabidopsis.org).

## Sequence and phylogenetic analysis

We used the ClustalW program to perform multiple alignments of TCP protein sequences from *B. juncea* var. tumida and *A. thaliana* (*Thompson et al., 1997*). A phylogenetic tree was constructed using MEGA 7.0 software and the maximum likelihood method based on the Poisson correction model and a bootstrap test replicated 1,000 times (*Tamura et al., 2013*). A gene structure diagram was drawn using the online software of the GSDS2.0 server (http://gsds.cbi.pku.edu.cn/) (*Hu et al., 2015*). The physical location data of *BjTCP* genes were retrieved from the *B. juncea* var. tumida genome. We subsequently mapped these *TCP* genes using MapInspect software. Conserved protein motifs were identified by using default parameters for the Multiple Em for Motif Elicitation (MEME; http://meme-suite.org/) program, and maximum 12 motifs were set. Subcellular localization of BjTCPs was predicted using ProtComp9.0 (http://www.softberry.com), and the identified protein motifs were further annotated using Weblogo (http://weblogo.berkeley.edu/). Finally, 2,000 bp of the 5′ sequence were used as the promoter region of each TCP gene to analyze the *cis*-acting elements using PlantCARE (http://bioinformatics.psb.ugent.be/webtools/plantcare/html/) (*Lescot et al., 2002*).

## Chromosomal location and prediction of *miR319* target genes

The physical location data of BjTCP genes were retrieved from the *B. juncea* var. tumida genomes. The mapping of these TCP genes was subsequently performed using MapInspect software. To predict miR target genes, we analyzed the full lengths of candidate TCP coding sequences using the psRNATarget website (*Dai & Zhao, 2011*).

## Expression profile of *TCP* genes

RNA-sequencing (RNA-seq) data from our previous research were downloaded from the National Center for Biotechnology Information Sequence Read Archive database (http://www.ncbi.nlm.nih.gov/sra/) with the following accession numbers: SRX108496 (Dayejie [DY] stems, a mutant variety without inflated stems, were collected 22 weeks after seeding), SRX108498 (YA1; Yonganxiaoye [YA] stems were collected 18 weeks after seeding), SRX108499 (YA2; YA stems were collected 20 weeks after seeding), SRX108500 (YA3; YA stems were collected 22 weeks after seeding),

SRX108501 (YA4; YA stems were collected 25 weeks after seeding), and SRX108502 (YAr; YA mix roots were collected 20 and 22 weeks after seeding) (*Sun et al., 2012*). Clean reads filtered from raw reads were mapped onto *B. juncea* genome version 1.5 (http://brassicadb.org/brad/datasets/pub/Genomes/Brassica_juncea/V1.5/) (*Yang et al., 2016*) using Tophat2 with default parameters (*Trapnell, Pachter & Salzberg, 2009*; *Trapnell et al., 2012*). Gene expression levels of individual genes were quantified using reads per kilobase of transcript per million (RPKM) values using Cufflinks 2.2.1 with default parameters (*Trapnell et al., 2012*).

### RNA extraction and real-time quantitative PCR analysis

Total RNA was extracted from different plant materials using RNA plant plus reagent (Tiangen Biotech Co., Ltd., Beijing, China) and treated with DNase I (Takara, Qingdao, China) to remove genomic DNA. Reverse transcription was performed using the Hiscript II 1st strand complementary DNA (cDNA) synthesis kit (Vazyme, Nanjing, China). Real-time quantitative reverse transcription polymerase chain reaction (qRT-PCR) was performed with 20 μL volume using TB Green™ *Premix Ex Taq*™ II (Tli RNaseH Plus) (Takara). *BjActin* was used as the internal reference gene for qRT-PCR; Table S1 lists gene-specific primers.

Three replicate samples of each period were subjected to three biological replicates using the BioRad IQ5 Real-Time PCR instrument (BioRad Laboratories, Hercules, CA, USA). Amplification parameters were as follows: activation at 50 °C for 2 min, predenaturation at 95 °C for 2 min, denaturation at 95 °C for 15 s, and annealing at 60 °C for 1 min for 40 cycles. Finally, the relative gene expression level was calculated using the $2^{-\Delta\Delta Ct}$ method (*Livak & Schmittgen, 2001*).

## RESULTS

### Identification of TCP family members in *B. juncea* var. tumida

To identify TCP proteins in *B. juncea* var. tumida, we screened out 63 genes and confirmed the domain using Pfam and Smart databases. Finally, we identified 62 *BjTCP* genes in *B. juncea* var. tumida. On the basis of similarity with *A. thaliana* homology genes, the 62 *BjTCP* genes were named with *BjTCP1a-BjTCP24d* (Table 1). The coding amino acids were from 171 to 639, with a molecular weight of 18.6–71.87 kDa and an isoelectric point (pI) of 5.5–10.18. Of the 62 genes, 61 were located on 18 chromosomes, except *BjTCP17b* anchored in contig6125. There was one TCP gene each on chromosomes A04, A08, A10, B01, and B06; two *TCP* genes each on chromosomes A01 and A05; and three to seven genes on other chromosomes (Figs. 1A–1R). We also found that most of the BjTCP proteins were localized in the nucleus, except BjTCP13a-c, whose location information was not found (Table 1), indicating that BjTCPs are TFs. These 62 TCP proteins may have multiple functions, and they mainly enriched in multiple GO terms, such as biological regulation, response to stimulus, rhythmic process and so on (Table S2).

### The phylogenetic tree of *BjTCP* genes and *AtTCP*s

Multiple-sequence alignment of TCP proteins showed that the conserved region was mainly focused on the TCP domain (Fig. S1).

**Table 1** The TCP protein family members in *B. juncea* var. tumida.

| ID | pfam domin (star-end) | | Name | chr | star | end | sence+/ antisence- | Subcellular locallzation | Homolog | PI | MW (kD) | protein (aa) |
|---|---|---|---|---|---|---|---|---|---|---|---|---|
| BjuA007230 | 84 | 242 | BjTCP1a | A02 | 10613454 | 10614491 | − | Nuclear | AtTCP1 | 5.5 | 39.28 | 346 |
| BjuA027377 | 84 | 232 | BjTCP1b | A07 | 29105069 | 29106097 | − | Nuclear | AtTCP1 | 6.68 | 38.96 | 343 |
| BjuB030534 | 84 | 233 | BjTCP1c | B03 | 29932727 | 29933758 | − | Nuclear | AtTCP1 | 5.98 | 39.1 | 344 |
| BjuB043984 | 85 | 235 | BjTCP1d | B03 | 27732982 | 27734025 | − | Nuclear | AtTCP1 | 5.96 | 39.57 | 348 |
| BjuB045720 | 81 | 300 | BjTCP1e | B05 | 38301640 | 38302668 | − | Nuclear | AtTCP1 | 5.88 | 38.91 | 343 |
| BjuB013551 | 72 | 251 | BjTCP2a | B05 | 13742319 | 13743080 | + | Nuclear | AtTCP2 | 6.64 | 27.45 | 254 |
| BjuB044682 | 161 | 261 | BjTCP2b | B02 | 58451263 | 58452048 | + | Nuclear | AtTCP2 | 6.71 | 28.53 | 262 |
| BjuA013153 | 162 | 259 | BjTCP2c | A01 | 16646189 | 16646968 | + | Nuclear | AtTCP2 | 6.69 | 28.34 | 260 |
| BjuA003953 | 81 | 236 | BjTCP2d | A01 | 5519388 | 5520104 | − | Nuclear | AtTCP2 | 6.48 | 25.82 | 259 |
| BjuB045012 | 1 | 158 | BjTCP3 | B07 | 27416073 | 27416864 | + | Nuclear | AtTCP3 | 6.55 | 28.64 | 264 |
| BjuB027204 | 34 | 231 | BjTCP5a | B04 | 184613 | 186371 | − | Nuclear | AtTCP5 | 9.33 | 30.31 | 275 |
| BjuB037760 | 63 | 364 | BjTCP5b | B02 | 59342015 | 59343112 | + | Nuclear | AtTCP5 | 6.21 | 40.73 | 365 |
| BjuB039698 | 62 | 262 | BjTCP5c | B07 | 12846872 | 12847951 | − | Nuclear | AtTCP5 | 6.95 | 34.36 | 304 |
| BjuA016046 | 36 | 211 | BjTCP6 | A04 | 13029721 | 13033965 | − | Nuclear | AtTCP6 | 7.98 | 71.87 | 639 |
| BjuA023481 | 42 | 223 | BjTCP7a | A06 | 20165661 | 20166410 | − | Nuclear | AtTCP7 | 9.69 | 27 | 250 |
| BjuA032507 | 14 | 189 | BjTCP7b | A09 | 5290320 | 5290967 | + | Nuclear | AtTCP7 | 9.38 | 23.12 | 216 |
| BjuA045586 | 1 | 151 | BjTCP7c | A02 | 34582679 | 34583206 | − | Nuclear | AtTCP7 | 7.92 | 18.6 | 176 |
| BjuB039713 | 14 | 190 | BjTCP7d | B07 | 12703979 | 12704632 | − | Nuclear | AtTCP7 | 9.82 | 23.3 | 218 |
| BjuB044801 | 42 | 235 | BjTCP7e | B02 | 60858548 | 60859336 | + | Nuclear | AtTCP7 | 9.51 | 28.37 | 263 |
| BjuA033567 | 53 | 218 | BjTCP8a | A09 | 15197090 | 15198274 | + | Nuclear | AtTCP8 | 6.09 | 41.43 | 395 |
| BjuB028163 | 54 | 232 | BjTCP8b | B04 | 9742480 | 9743682 | − | Nuclear | AtTCP8 | 6 | 42.26 | 401 |
| BjuA010736 | 71 | 197 | BjTCP9a | A03 | 13517323 | 13518273 | − | Nuclear | AtTCP9 | 9.86 | 33.84 | 317 |
| BjuA018039 | 62 | 178 | BjTCP9b | A05 | 3386569 | 3387543 | − | Nuclear | AtTCP9 | 9.41 | 33.43 | 325 |
| BjuB016827 | 72 | 201 | BjTCP9c | B08 | 26586920 | 26587915 | − | Nuclear | AtTCP9 | 9.67 | 35.15 | 332 |
| BjuB020003 | 67 | 184 | BjTCP9d | B06 | 3171113 | 3172111 | − | Nuclear | AtTCP9 | 9.58 | 35.15 | 333 |
| BjuA041558 | 100 | 193 | BjTCP12a | A02 | 11715044 | 11716146 | − | Nuclear | AtTCP12 | 8.78 | 37.88 | 322 |
| BjuB010789 | 103 | 231 | BjTCP12b | B05 | 55023718 | 55024870 | − | Nuclear | AtTCP12 | 8.45 | 39.61 | 352 |
| BjuA011472 | 59 | 220 | BjTCP13a | A03 | 18170796 | 18171755 | + | − | AtTCP13 | 6.73 | 35.69 | 320 |
| BjuA021096 | 86 | 243 | BjTCP13b | A05 | 31265695 | 31266872 | − | – | AtTCP13 | 6.63 | 35.72 | 321 |
| BjuB026804 | 58 | 214 | BjTCP13c | B07 | 14926766 | 14927692 | − | – | AtTCP13 | 7.94 | 34.47 | 309 |
| BjuA022523 | 1 | 178 | BjTCP14a | A06 | 12060176 | 12061564 | − | Nuclear | AtTCP14 | 5.98 | 36.45 | 344 |
| BjuB019669 | 103 | 292 | BjTCP14b | B08 | 54134488 | 54135858 | − | Nuclear | AtTCP14 | 6.48 | 49.33 | 457 |
| BjuA007311 | 58 | 201 | BjTCP15a | A02 | 12233735 | 12234697 | + | Nuclear | AtTCP15 | 6.91 | 33.84 | 321 |
| BjuA016487 | 55 | 195 | BjTCP15b | A07 | 30212223 | 30213185 | + | Nuclear | AtTCP15 | 6.91 | 34.06 | 321 |
| BjuA027170 | 56 | 197 | BjTCP15c | A07 | 27540133 | 27541098 | − | Nuclear | AtTCP15 | 7.43 | 33.96 | 322 |
| BjuB000709 | 50 | 186 | BjTCP15d | B05 | 51941921 | 51942820 | − | Nuclear | AtTCP15 | 8.05 | 32.02 | 300 |
| BjuB003932 | 57 | 194 | BjTCP15e | B03 | 29008395 | 29009351 | + | Nuclear | AtTCP15 | 7.43 | 33.74 | 319 |
| BjuB030482 | 55 | 196 | BjTCP15f | B03 | 32626814 | 32627776 | + | Nuclear | AtTCP15 | 7.15 | 33.49 | 321 |
| BjuA009092 | 32 | 164 | BjTCP17a | A03 | 1624898 | 1625620 | + | Nuclear | AtTCP17 | 8.53 | 19.12 | 171 |
| BjuO008355 | 37 | 254 | BjTCP17b | Contig6125 | 66534 | 67304 | + | Nuclear | AtTCP17 | 6.7 | 28.54 | 257 |

| ID | pfam domin (star-end) | | Name | chr | star | end | sence+/ antisence- | Subcellular locallzation | Homolog | PI | MW (kD) | protein (aa) |
|---|---|---|---|---|---|---|---|---|---|---|---|---|
| BjuA012606 | 154 | 319 | BjTCP18a | A03 | 22282211 | 22283992 | + | Nuclear | AtTCP18 | 8.56 | 48.45 | 425 |
| BjuB007175 | 150 | 311 | BjTCP18b | B07 | 5163145 | 5165165 | + | Nuclear | AtTCP18 | 7.32 | 40.12 | 350 |
| BjuB007177 | 150 | 311 | BjTCP18c | B07 | 5138717 | 5140787 | + | Nuclear | AtTCP18 | 6.66 | 50.08 | 437 |
| BjuB025473 | 159 | 292 | BjTCP18d | B01 | 38022785 | 38024462 | − | Nuclear | AtTCP18 | 8.4 | 46.6 | 404 |
| BjuA007026 | 52 | 174 | BjTCP19 | A02 | 9082363 | 9083205 | + | Nuclear | AtTCP19 | 5.5 | 30.18 | 281 |
| BjuA024339 | 67 | 304 | BjTCP20a | A06 | 25596111 | 25597025 | + | Nuclear | AtTCP20 | 7.97 | 32.22 | 305 |
| BjuA031722 | 53 | 239 | BjTCP20b | A09 | 3333673 | 3334532 | − | Nuclear | AtTCP20 | 5.17 | 25.25 | 241 |
| BjuB037176 | 65 | 309 | BjTCP20c | B02 | 56079378 | 56080310 | − | Nuclear | AtTCP20 | 7.3 | 32.61 | 311 |
| BjuA009108 | 31 | 206 | BjTCP21a | A03 | 1689417 | 1690118 | + | Nuclear | AtTCP21 | 10.18 | 24.26 | 234 |
| BjuA041017 | 32 | 207 | BjTCP21b | A02 | 1466529 | 1467236 | + | Nuclear | AtTCP21 | 9.29 | 24.54 | 236 |
| BjuA047338 | 31 | 208 | BjTCP21c | A10 | 17491637 | 17492344 | + | Nuclear | AtTCP21 | 7.99 | 24.73 | 236 |
| BjuB012430 | 31 | 209 | BjTCP21d | B05 | 17001527 | 17002243 | − | Nuclear | AtTCP21 | 9.57 | 25.05 | 239 |
| BjuB040955 | 33 | 209 | BjTCP21e | B08 | 2452006 | 2452713 | + | Nuclear | AtTCP21 | 10.18 | 24.46 | 236 |
| BjuB048495 | 28 | 209 | BjTCP21f | B02 | 50550592 | 50551305 | − | Nuclear | AtTCP21 | 7.99 | 24.94 | 238 |
| BjuA007449 | 57 | 190 | BjTCP22a | A02 | 13365484 | 13366605 | + | Nuclear | AtTCP22,AtTCP23 | 8.63 | 39.06 | 374 |
| BjuA043373 | 40 | 202 | BjTCP22b | A07 | 31378278 | 31379348 | + | Nuclear | AtTCP22,AtTCP23 | 6.87 | 37.34 | 357 |
| BjuB010697 | 56 | 186 | BjTCP22c | B05 | 59790473 | 59791573 | − | Nuclear | AtTCP22,AtTCP23 | 8.63 | 38.41 | 367 |
| BjuB044035 | 45 | 168 | BjTCP22d | B03 | 38473563 | 38474609 | + | Nuclear | AtTCP22,AtTCP23 | 6.31 | 36.59 | 349 |
| BjuA034777 | 46 | 138 | BjTCP24a | A09 | 31311472 | 31312425 | + | Nuclear | AtTCP24 | 7.8 | 35.29 | 318 |
| BjuB029526 | 47 | 139 | BjTCP24b | B04 | 35033858 | 35034823 | − | Nuclear | AtTCP24 | 6.90 | 35.7 | 322 |
| BjuB032913 | 66 | 306 | BjTCP24c | B03 | 5929399 | 5930376 | + | Nuclear | AtTCP24 | 7.16 | 36.57 | 326 |
| BjuA029872 | 55 | 284 | BjTCP24d | A08 | 20433785 | 20434732 | + | Nuclear | AtTCP24 | 6.81 | 35.46 | 316 |

To assess the phylogenetic relationships of the TCP family, we used the predicted TCP protein sequences from *B. juncea* var. tumida and *A. thaliana* to construct a phylogenetic tree. Results indicated that all TCP proteins are divided into two groups, class I and class II (Fig. 2A). In class II, the TCP proteins were further subdivided into CYC, TB1, and CIN groups. The CYC group was mainly clustered by AtTCP1 and AtTCP12, containing four AtTCP1 homologous proteins BjTCP1a-d, BjTCP2a-d, and two AtTCP12 homologous proteins BjTCP12a-b. The TB1 group comprised AtTCP18 and four homologous TCP proteins BjTCP18a-d. In the CIN group, we found no proteins to be homologous with AtTCP4 and AtTCP10, while the other TCP proteins had at least one homologous protein, such as AtTCP24 (four homologous proteins BjTCP24a–d), AtTCP13 (three homologous proteins BjTCP13a–c), AtTCP17 (two homology proteins BjTCP17a and BjTCP17b), and AtTCP5 (three homologous proteins BjTCP5a–c). In class I, we found no homologous proteins in *B. juncea* var. tumida, except AtTCP11, AtTCP16, and AtTCP23, but the other TCP proteins had multiple homologous proteins, such as AtTCP15, and AtTCP21 even had six homologous proteins.

Interestingly, a series of genes, such as *BjTCP15b*, *BjTCP15c*, *BjTCP1b*, and *BjTCP22b*, were located on the same chromosome A07 (Fig. 1G). Their homologous genes (*BjTCP15e*, *BjTCP15f, BjTCP1c*, and *BjTCP22d*) showed the same order on chromosome B03 (Figs. 1M

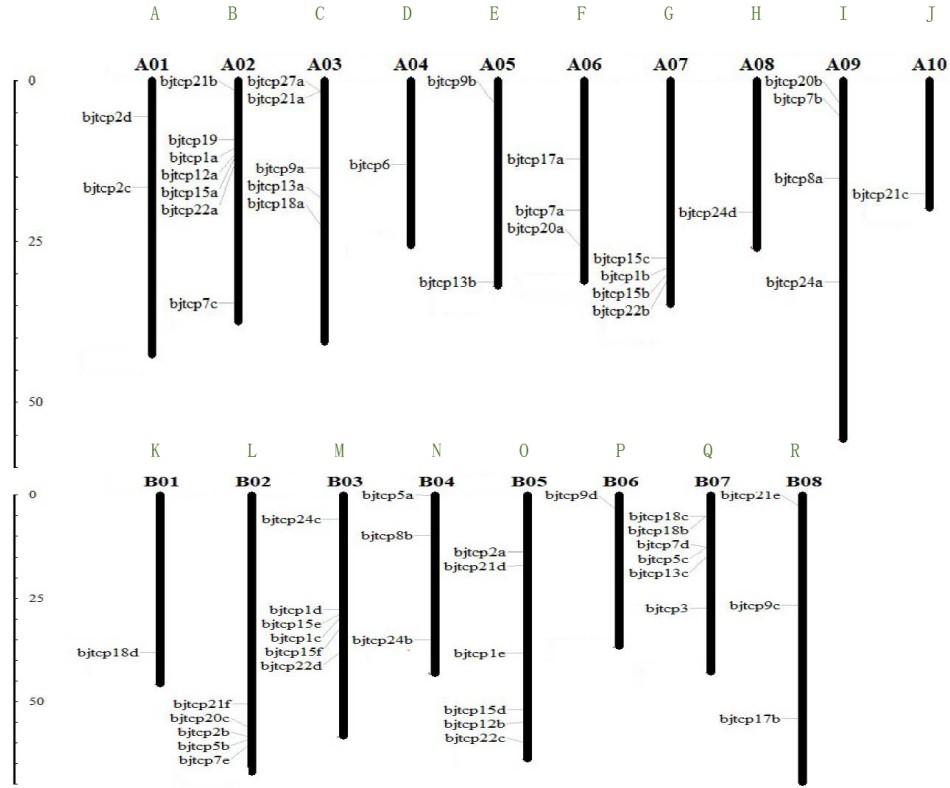

**Figure 1 The gene locations of *BjTCP* gene family.** The chromosome name is at the top of each bar. The scale of the chromosome is in millions of bases (Mb).

and 2). The eight genes were searched in *B. rapa* (AA) and *B. nigra* (BB) using the BLASTP program, and four highly similar genes were screened out in *B. rapa* (AA) and *B. nigra* (BB). The evolutionary relationships suggested that the four genes located on chromosome A07 were clustered with the homologous genes of a subgenome ancestor *B. rapa* (AA) in one branch (Fig. S2). The four genes on chromosome B03 also corresponded to the B subgenome ancestor *B. nigra* (BB) (Fig. S2). These results indicated that the fragments between the four genes of chromosomes A07 and B03 might be formed from *B. rapa* (AA) and *B. nigra* (BB), respectively.

BjTCP proteins had a typical bHLH motif in all identified TCP proteins (Figs. 2B, 2C). In *A. thaliana*, the main difference between classes I and II was the identity of the residue at positions 10–15 of the TCP domain. Most class I BjTCP proteins lost four amino acids at positions 9–13 and had Gly at position 15, while class II BjTCP proteins had Asp at position 15 in the TCP domain (Figs. 2B, 2C).

## Gene structures and conserved motif analysis of *BjTCP* genes

To further analyze the characteristic of *BiTCP* genes, we explored the exon/intron gene structure. Results indicated that most *BjTCP* genes only have one exon, except *BjTCP18s*, *BjTCP12s*, *BjTCP20b*, and *BjTCP13b*, which contain two or more exons. We also found that

A

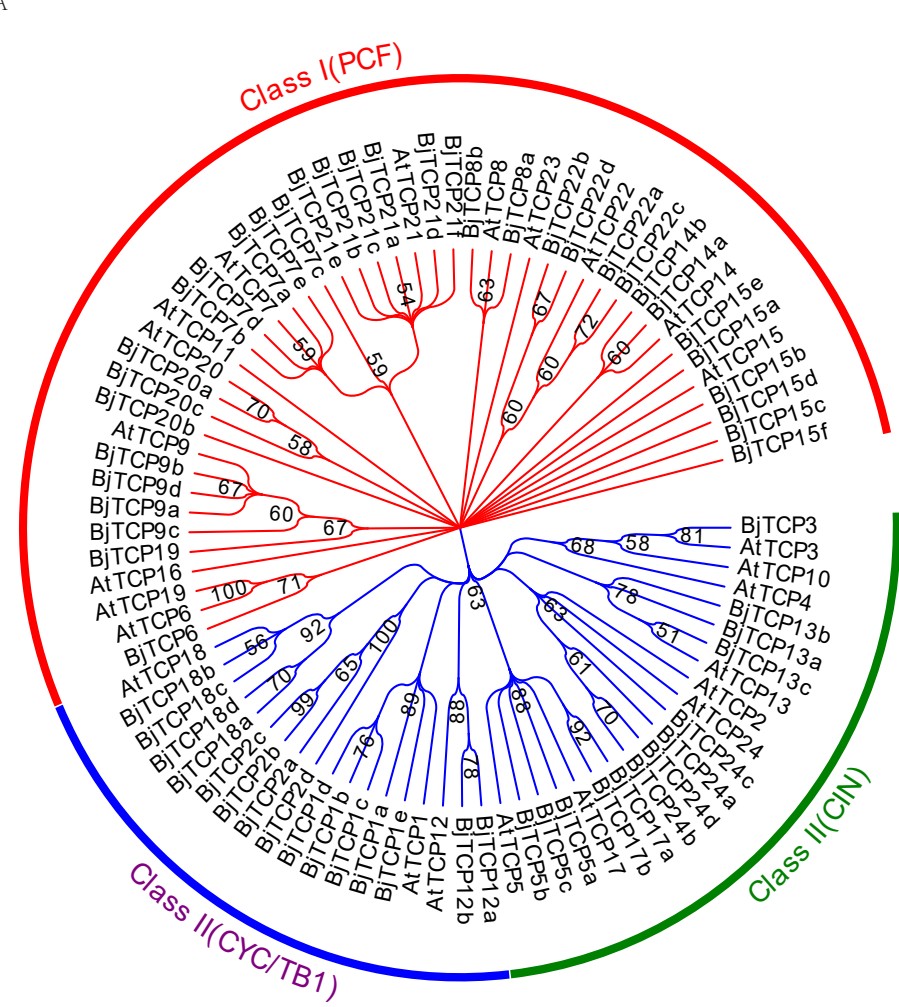

B

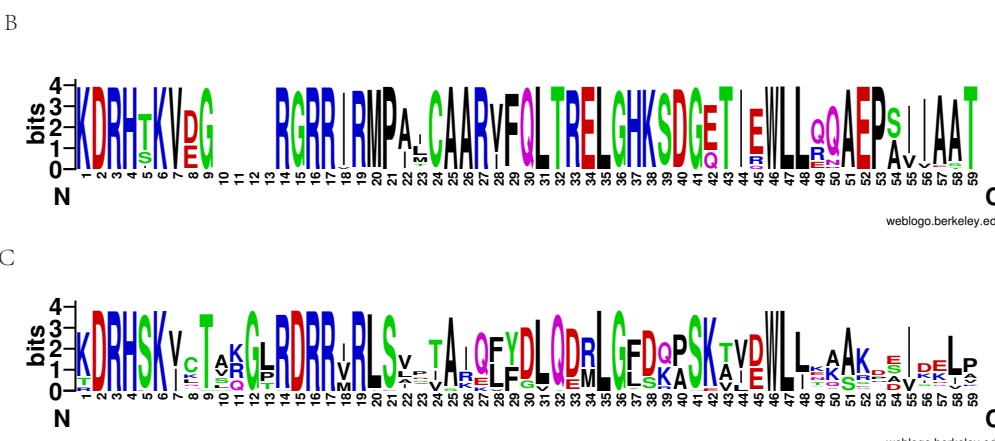

C

**Figure 2 Evolutionary relationships of taxa.** (A) The evolutionary history was inferred by using the Maximum Likelihood method based on the Poisson correction model. The bootstrap consensus tree inferred from 1,000 replicates is taken to represent the evolutionary history 

**Figure 2 (…continued)**
of the taxa analyzed. Branches corresponding to partitions reproduced in less than 50% bootstrap replicates are collapsed. Initial tree(s) for the heuristic search were obtained automatically by applying Neighbor-Join and BioNJ algorithms to a matrix of pairwise distances estimated using a JTT model, and then selecting the topology with superior log likelihood value. The analysis involved 86 amino acid sequences. All positions containing gaps and missing data were eliminated. There were a total of 66 positions in the final dataset. Evolutionary analyses were conducted in MEGA7. (B) A conserved motif in the Class I subfamily of the *BjTCP* gene family. (C) A conserved motif in the Class II subfamily of the BjTCP gene family. The consensus sequences were displayed using Weblogo (http://weblogo.berkeley.edu).

the genetic structure and evolutionary relationships of all TCP family members of *B. juncea* var. tumida are closely related. Genes within the same subfamily often showed similar gene structures. *BjTCP12a* and *BjTCP12b* comprised two exons, while *BjTCP18a-d* comprised more than three exons (Figs. 3A, 3B). In Chinese cabbage, the homologous *BrTCP1a* and *BrTCP1b* genes have two exons, and this exon number was highly similar to *B. juncea* var. tumida's homology genes; for example, *BrTCP6–13* and *BrTCP15* had only one exon. Although *BjTCP18a* and *BjTCP18b* have two exons, *BjTCP18c* and *BjTCP18d* have four exons, which is similar with their homologous *B. rapa* *BrTCP18a* and *BrTCP18b* genes (*Liu et al., 2018b*). The conserved motifs in these *BjTCP* genes also showed similar characters within the same subgroup, such as three similar motifs in all BjTCP15 homologous proteins and five similar motifs in all BjTCP21 homologous proteins (Fig. 3C).

## *BjTCP* genes with miR319 target sites

In *Arabidopsis*, *AtTCP2-4*, *AtTCP10*, and *AtTCP24* are post-transcriptionally regulated by miR319 (*Bresso et al., 2018*; *Palatnik et al., 2003*). In *B. juncea* var. tumida, the evolutionarily closest homologs of these genes are *BjTCP2a–d*, and *BjTCP24a–d*, which contain sequences well matched with miR319 and might be the targets of miRs (Fig. 4). *BjTCP3* did not contain the putative miR319 recognition site, but mismatches of other genes mainly existed at 3′ of miR319 and 5′ of the targeted *BjTCP* mRNA, and core target sequences were conserved.

## Promoter *cis*-acting element analysis of *BjTCP* genes

The *cis*-acting elements in the promoter of a gene usually regulate gene expression and function. In this study, we found multiple *cis*-acting elements in TCP gene promoters, such as plant hormone response elements, light response elements, stress response elements, meristem expression, circadian control, and low-temperature and wound response elements (Fig. 5 and Table S3).

For hormone-related *cis*-acting elements, we identified abscisic acid (ABA) response elements (ABREs) and found at least two or more ABRE *cis*-acting elements in *B. juncea* var. tumida TCP gene promoters, expect *BjTCP5s*, *BjTCP18s*, *BjTCP19*, and *BjTCP24*. Auxin response elements included AuxRP and TGA elements, and AuxRP had a relatively small number of components, mainly in the *BjTCP5*, *BjTCP19*, and *BjTCP20* promoters, while the TGA element was relatively more extensively distributed. The MeJA response elements CGTCA and TGACG were found on most promoters, except for *BjTCP12*, *BjTCP19*, and *BjTCP20*. We also found a number of other hormone-related *cis*-elements, such as ethylene

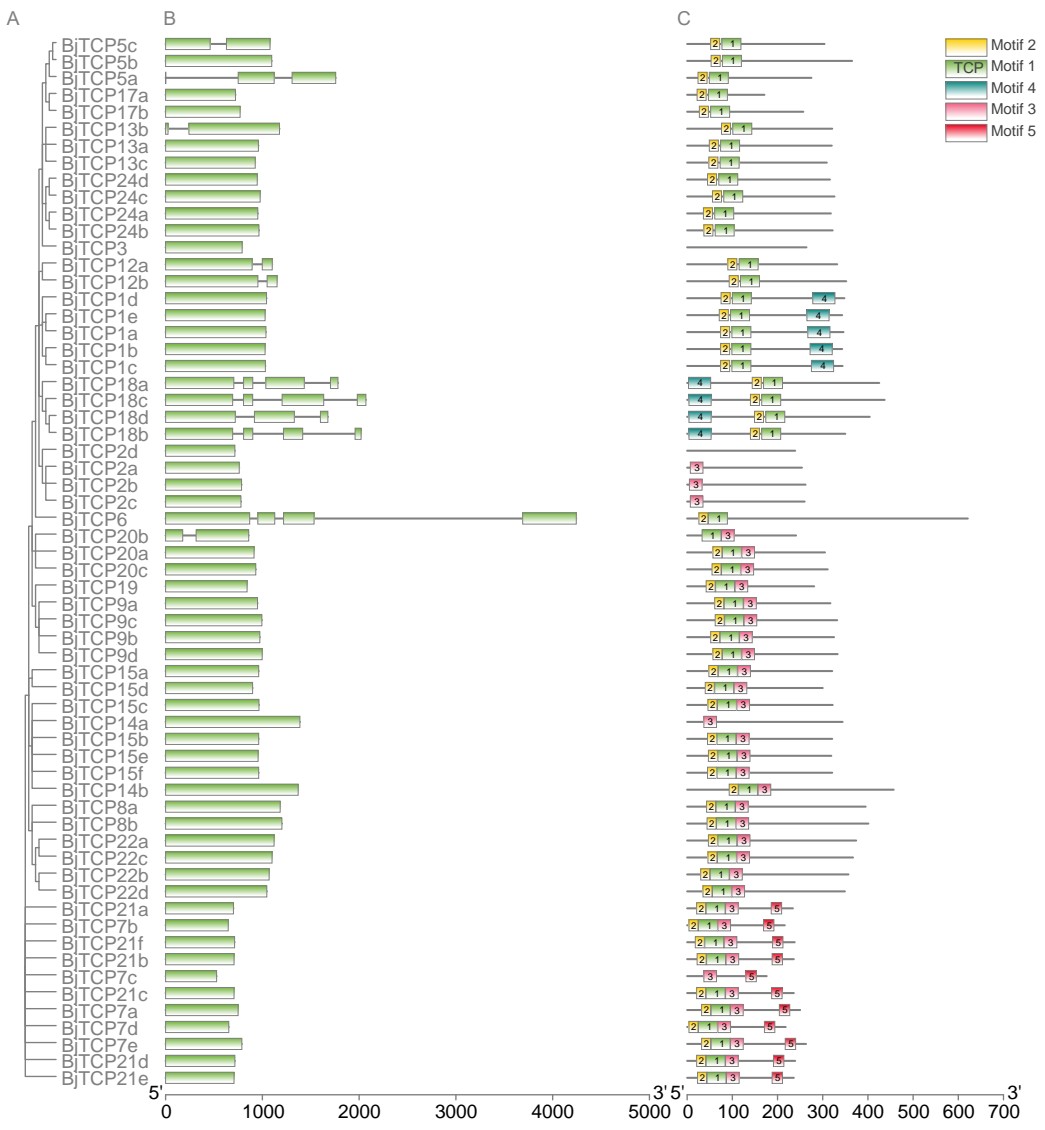

**Figure 3** **Genomic structure and motif composition of BjTCPs.** (A) The phylogenetic tree of BjTCP proteins. (B) Genomic structure of *BjTCPs* family members in tumorous stem mustard. Exons and introns are represented with blank boxes and blank lines. (C) The conserved motifs in tumorous stem mustard TCP proteins were identified using MEME. Each motif is represented with a specific color and the characters sequence were showed below.

(ET) response element (ERE), GA response elements (GAREs) P-box and TATC-box, and the SA response element TCA element in some *BjTCP* promoters.

In addition, we found a large number of *cis*-acting elements related to light response in these promoters, including the 3-AF1 binding site, ACE, AE-box, TCT-motif, ATC-motif, Box 4, GATA-motif, G-Box, GT1-motif, and I-box. We also found other elements, including WUN-motif (related to wounds), meristem element (related to the meristem), circadian element (related to circadian control), LTR element (related to low-temperature induction),

```
miR319     U C C C U C G A G G G A A G U C A G G U U
BjTCP2a    C A G G G G G A C C C U U C A G U C C A A
BjTCP2b    C A G G G G G A C C C U U C A G U C C A A
BjTCP2c    C A G G G G G A C C C U U C A G U C C A A
BjTCP2d    C A G G G G G A C C C U U C A G U C C A A
BjTCP24a   U A G G G G G A C C C U U C A G U C C A A
BjTCP24b   U A G G G G G A C C C U U C A G U C C A A
BjTCP24c   U A G G G G G A C C C U U C A G U C C A A
BjTCP24d   U A G G G G G A C C C U U C A G U C C A A
```

**Figure 4  Alignment of putative target areas for miR319.** Mismatches and G-U wobbles were represented by yellow and green, respectively.

and defense and stress responsiveness elements (including MBS, Myb, Myc, STRE, TC-rich, W box, and ARE elements). In particular, we identified MYB and MYC-motif elements in almost all TCP promoters.

## Tissue-specific expression profiles of *BjTCP* genes

In *A. thaliana*, TCP proteins were found to be mainly involved in development and defense. We analyzed the expression patterns of all *TCP* genes in different development periods and tissues on the basis of previous RNA-seq data (*Sun et al., 2012*). *BjTCP1a* and *BjTCP5c* expression could not be detected in all samples. However, 26 *TCP* genes were highly expressed in at least two tissues (log2 (fragments per kilobase of transcript per million [FPKM]) ≥ 3; subgroup half-bottom in Fig. 6). *BjTCP13b* and *BjTCP5a* were weakly expressed in no-swelling strain (DA) samples. We did not detect *BjTCP1b*, *BjTCP1c*, *BjTCP1d*, *BjTCP5c*, *BjTCP7c*, *BjTCP13a*, *BjTCP13c*, *BjTCP18d*, and *BjTCP19* expression in DY stem tissue. *BjTCP12a*, *BjTCP12b* and *BjTCP18c* were weakly expressed in DY, YA1, and/or YA2 strains (Fig. 6).

In addition, we analyzed the expression profiles of *B. juncea* var. tumida seedlings and tumorous stems. The expression levels of these four genes (*BjTCP18a-d*) were low in these tissues; the expression levels gradually decreased along with the swelling of tumorous stems (Fig. S3).

## Expression analysis of *BjTCP* genes in response to exogenous hormones

To predict the possible functions of *TCP* genes in environmental adaptation, we investigated their transcriptional profile after SA and GA treatment. Multiple gene family members in the same branch often have highly similar sequence characteristics and contain similar *cis*-acting elements (Fig. 5), so we selected one corresponding homologous gene from each *A. thaliana* TCP for qRT-PCR analysis.

After SA treatment for 2–8 h, almost all the detected *BjTCP* genes were upregulated, except *BjTCP1a*, *BjTCP12a*, and *BjTCP17a*, while after 24 h, the *BjTCP* genes were downregulated to a low level. *BjTCP9a*, *BjTCP13a*, *BjTCP15a*, *BjTCP18a*, *BjTCP19a*,
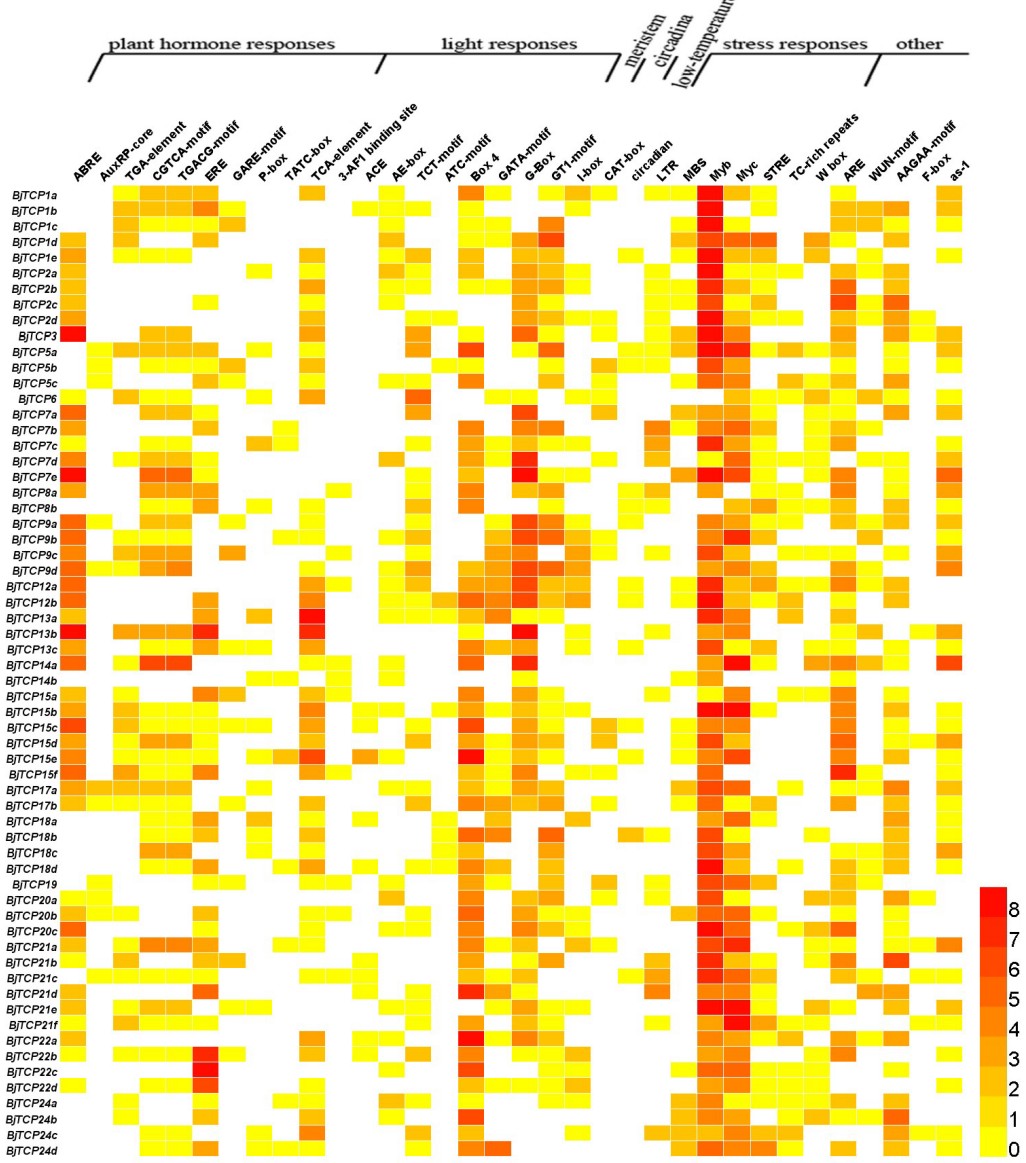

**Figure 5** **Cis-acting elements on promoters of *BjTCP* genes.** The colour bar shows the number of cis-acting elements.

*BjTCP22a*, and *BjTCP24a* were induced at early stages and maintained at a relatively high level until 8 h. *BjTCP17a* was induced slowly and was highly expressed 24 h after SA treatment (Fig. 7).

After GA treatment, we did not detect *BjTCP12a, BjTCP17a*, and *BjTCP20a* expression. In contrast, the expression of other genes was induced at early stages but decreased to a low level mainly after 8 h (Fig. 7).

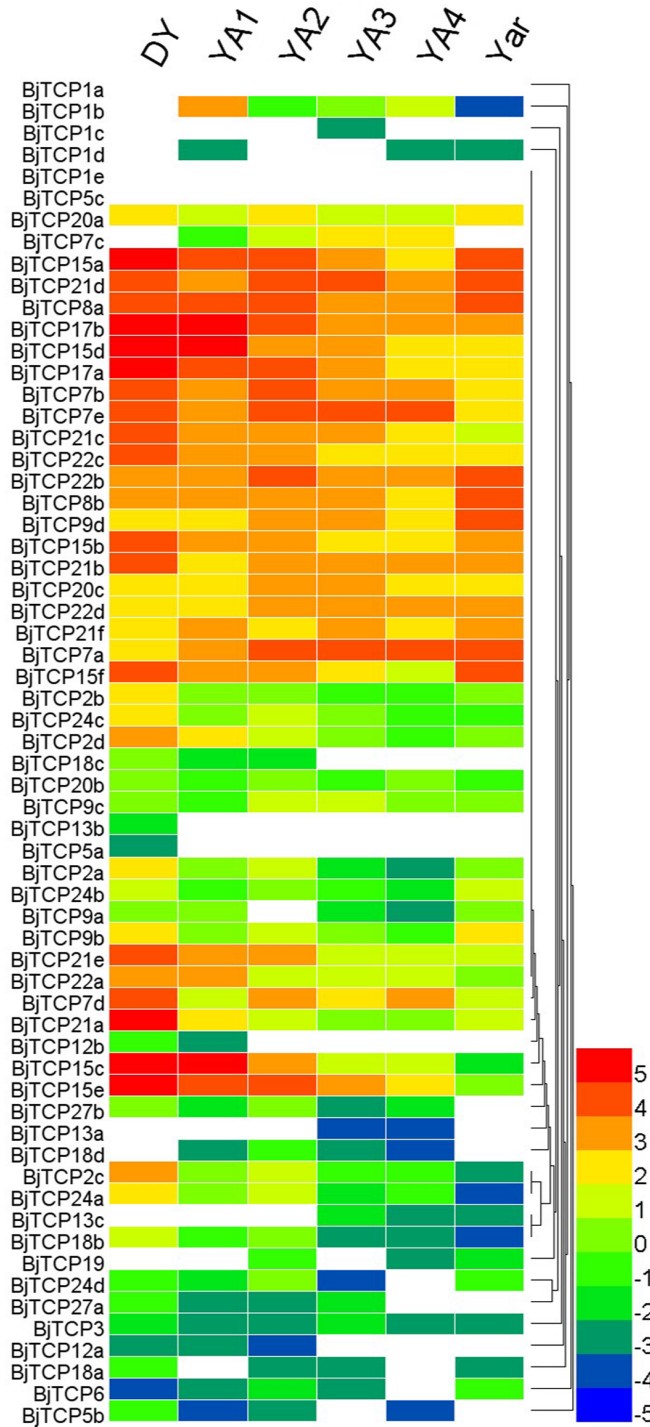

**Figure 6** **Expression patterns of TCP genes in different tissues and development stages of *B. juncea* var. tumida.** DY, *Dayejie* stems were collected 22 weeks after seeding (daye3bianzhong); YA1-4, The stems of *Yong'an* were collected 18, 20, 22, and 25 weeks after seeding; YAr, The mix roots samples of 18 and 22 weeks after seeding. The expression levels are represented by the color bar (log2-transformed).

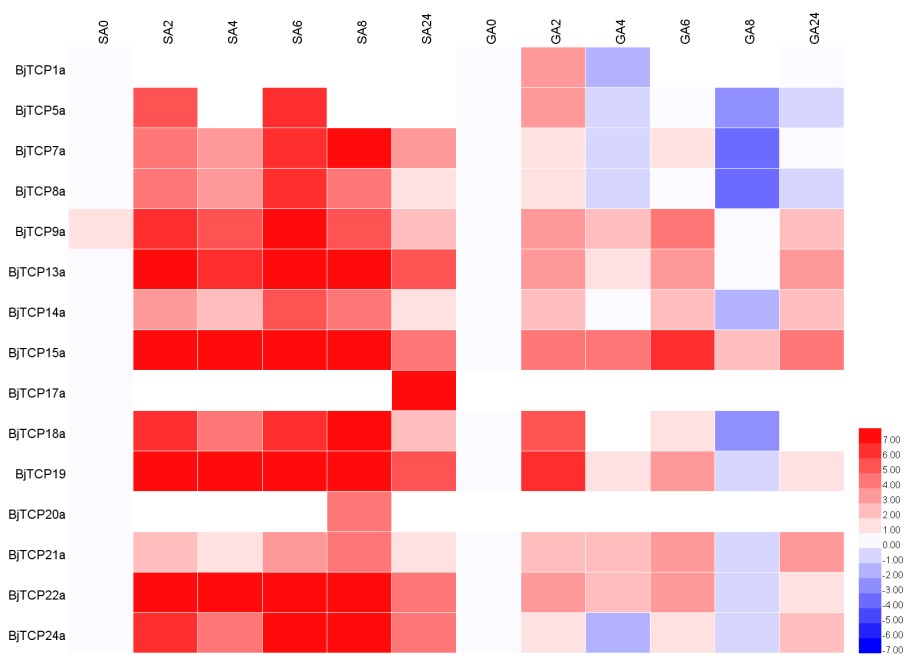

**Figure 7** **Expression levels of *BjTCPs* under SA and GA treatment by qRT-PCR.** The number represented the treatment times (hours). The colour scales represent relative expression data.

# DISCUSSION

Plant-specific TCP TFs play various roles in plant growth and development. In many plants, the general organization of the TCP family is conserved, and there are more members in class I compared to class II (*Du et al., 2017*; *Li et al., 2017*; *Liu et al., 2018b*; *Ma et al., 2014*; *Ma et al., 2016*; *Shi et al., 2016*; *Wang et al., 2018*; *Wang et al., 2019c*; *Zhao et al., 2018*; *Zheng et al., 2018*). In *B. juncea* var. tumida, we found 62 *BjTCP* genes. As a tetraploid plant, *B. juncea* var. tumida contains twice as many TCP proteins as *Arabidopsis* (24 TCP proteins), indicating that some genes are duplicated during evolution. However, recent studies have reported 36 and 38 *TCP* genes in *B. rapa* (AA) and *B. oleracea* (CC), respectively (*Liu et al., 2019*), indicating that the *B. juncea* genome might contain >70 *TCP* genes, although we identified only 62. We used HMMER 3.0 to search for TCP domain proteins and further removed members that did not contain the TCP domain using the Pfam database. We did not find the homologous genes of *AtTCP4*, *AtTCP10*, *AtTCP11*, and *AtTCP16*, which is similar to the study by *Liu et al. (2018b)*, who did not find the homologous genes of *ATCP11* and *AtTCP16* during the identification of the TCP family in the Chinese cabbage using the BLASTP program (*Liu et al., 2018b*). In this study, *BjuA003953*, *BjuA026354*, *BjuB013551*, and *BjuB044682* (named *BjTCP2a–d*) were found to be similar to *AtTCP2*, but the confidence level was low when analyzed using the Pfam database, and MEME analysis showed that no motif included the TCP domain. In addition, the *BjTCP2a-d* amino acid sequence was not clustered with the homolog protein AtTCP2

in the phylogenetic tree, which might be because of the incompletely predicted amino acid sequences during assembling of the genome sequence.

The exon/intron gene structure, conserved motif distribution patterns, and *BjTCP* homologous gene domain often show high similarity, such as *BjTCP21a-f*, *BjTCP12a/b*, and so on, and we believe that these similarities within the cluster of homologous genes members suggest that they might have a similar function during *B. juncea* var. tumida growth and development.

We located two genes clusters (*BjTCP15b*, *BjTCP15c*, *BjTCP1b*, and *BjTCP22b*) and (*BjTCP15f*, *BjTCP15e*, *BjTCP1c*, and *BjTCP22d*) on chromosomes A07 and B03, respectively. Evolutionary relationship results showed the allopolyploid *B. juncea* var. tumida (*B.juncea*, AABB) might form by hybridization between the diploid ancestors of *B. rapa* (AA) and *B. nigra* (BB), followed by spontaneous chromosome doubling. These results also indicated that the division of *BjTCP15b* and *BjTCP15c* might occur earlier than tetraploid formation.

As mentioned before, there are 24 TCP genes in *Arabidopsis*. Some corresponding homologous of TCP genes were not found in *B. juncea* var. tumida, such as *TCP4*, *TCP10*, *TCP11*, and *TCP16*, probably due to gene loss events during evolution. *B. juncea* var. tumida is a tetraploid plant that belongs to the cruciferous near-source species of *Arabidopsis*. In *B. juncea* var. tumida, some *TCP* genes have more than two homologous genes, such as *BjTCP1* (five homologous genes), *BjTCP18* (four homologous genes), *BjTCP21* (six homologous genes), and *BjTCP15* (six homologous genes). These genes may be formed by multiple gene duplication events, and the functions of these paralogous genes gradually differentiated during evolution. Most paralogous genes had similar *cis*-acting elements, but there were a few differences. For example, the four paralogues of *BjTCP18* had no ABA and auxin *cis*-acting elements, but *BjTCP18b* was the only member with circadian regulatory elements, suggesting that *BjTCP18b* might be involved in the circadian rhythm. Correspondingly, the expression patterns of these paralogous genes were also different.

In addition, six homologous genes of *AtTCP15* and *AtTCP21* in *B. juncea* var. tumida are interesting. *AtTCP15* plays an important role in regulating endoreduplication during development in *Arabidopsis* (*Li, Li & Dong, 2012*; *Uberti-Manassero et al., 2012*). In different developmental stages, the six *BjTCP* genes are highly expressed in DY, but there are chronological differences among the swollen tuber cultivars, suggesting that several genes might be involved in developmental regulation at different stages.

Interestingly, as mentioned before, in *Arabidopsis*, *AtTCP2*, *AtTCP3*, *AtTCP4*, *AtTCP10*, and *AtTCP24* are post-transcriptionally regulated by miR319, and these genes mainly regulate leaf morphogenesis and senescence (*Bresso et al., 2018*; *Palatnik et al., 2003*). In *B. juncea* var. tumida, no *AtTCP4* and *AtTCP10* homolog genes have been identified, and *BjTCP3* does not contain the miR319 regulation site. Only *BjTCP2a–d* and *BjTCP24a–d* have the putative miR319 recognition site, and their expression levels in stem development of the swelling strains (YA1–YA4) are relatively low compared to the no-swelling strain (DA). These results indicate that miR319 might not be involved in stem-swelling regulation in *B. juncea* var. tumida.

The *Arabidopsis BRANCHED1* (*BRC1*), the rice *TB1*, and the maize *TB1* function as negative regulators of the growth of axillary buds and branching (*Aguilar-Martinez, Poza-Carrion & Cubas, 2007*; *Dixon et al., 2018*; *Finlayson, 2007*; *Takeda et al., 2003*). *BRC1/TB1* orthologues play a similar role in the development of the primary shoot architecture and negatively regulate lateral branching (*Aguilar-Martinez, Poza-Carrion & Cubas, 2007*; *Dixon et al., 2018*; *Finlayson, 2007*; *Gonzalez-Grandio et al., 2013*; *Muhr et al., 2016*; *Wang et al., 2019a*; *Yang et al., 2015*). In addition, *OsTB1* can be regulated by *IPA1* to suppress tillering in rice, and *TB1* can interact with *FT1* to regulate inflorescence architecture in bread wheat (*Dixon et al., 2018*; *Guo et al., 2013*; *Takeda et al., 2003*). In *B. juncea* var. tumida, which has a close phylogenetic relationship with *Arabidopsis*, *BjTCP18* s might play a similar function in branching. There are four *BjTCP18* homologous genes in *B. juncea* var. tumida, which might have been formed by gene duplication. Functional differentiation might occur between the four *TCP18* genes, given their differential expression patterns during tissue development. The flowering stage in *B. juncea* var. tumida is mainly characterized by swelling of the tumorous stem. At this time, the plant shows a bolting and flowering phenomenon similar to *Arabidopsis*. Since *BRC1* inhibits branching and flowering, gradual downregulation of its messenger RNA (mRNA) levels might reflect a gradual decrease in the ability to inhibit branching and flowering. These events also indicate that *B. juncea* var. tumida is about to enter the period of reproductive growth.

There are 16 varieties of mustard species identified and used for food consumption, in which the main difference is the tissue shape, including the root, stem, leaf, and branch (*Qiao, Liu & Lei, 1998*). The *BRC1* gene controls plant branching and interacts with the flowering time–relate gene *FT*, and four identified *BjBRC1* genes might imply further functional differentiation of branch development and floral transition. Among these *BjTCP* genes, multiple *BjTCP15* and *BjTCP21* genes are highly expressed in the DY and/or the early stage of seedling and tumorous stem per-swelling stage as compared to the swelling stage. DY is a mutant line with no swelling, and YA1 and YA2 also have not started to swell. These findings indicate that these genes are involved in the process of stem swelling in *B. juncea* var. tumida.

Increasing evidence verifies that TCP proteins are involved in responses to plant hormones (*Braun et al., 2012*; *Danisman et al., 2012*; *Dun et al., 2012*; *Feng et al., 2018*; *Gonzalez-Grandio et al., 2017*; *Hay, Barkoulas & Tsiantis, 2004*; *He et al., 2016*; *Liu et al., 2018a*; *Lopez et al., 2015*; *Nicolas & Cubas, 2016*; *Qin et al., 2005*; *Schommer et al., 2008*; *Shen et al., 2019*; *Wang et al., 2019a*; *Wang et al., 2013*). In this study, most of *B. juncea* var. tumida *TCP* genes appeared to be regulated by SA and GA. In *A. thaliana*, several TCPs interact with the SA biosynthetic enzyme *ISOCHORISMATE SYNTHASE 1* gene and enhance its expression by binding to the TCP-binding motif in its promoter region (*Wang et al., 2015*). Our results showed that there are many SA-related *cis*-elements in the promoter regions of *BjTCP* genes, and the expression levels of several *BjTCP* genes significantly increase after SA treatment, indicating that *BjTCP* genes might be involved in SA signal transduction. However, SA treatment does not seem to directly affect the expression of these genes. For example, the promoter regions of *BjTCP1a* and *BjTCP12a* contain two TCA elements, but there was almost no expression of these genes. In contrast,

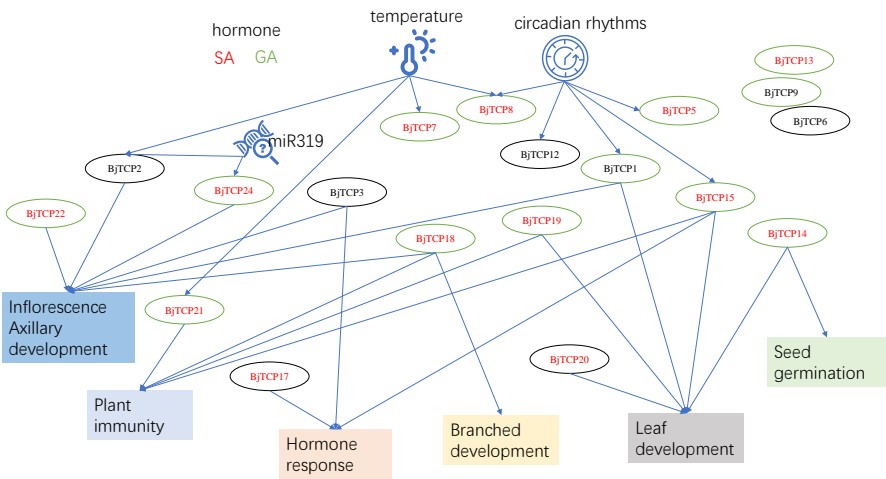

**Figure 8** **The putative mechanism diagram on the basis of current results and the reports of proximal species *A. thaliana*.** Red letter shows the genes may regulated by SA. A green circle means the genes may induced by GA. Arrows indicate possible regulatory relationships.

although the promoter regions of *BjTCP7a*, *BjTCP8a*, *BjTCP17a*, and *BjTCP20a* contain no TCA elements, the expression levels of these four genes changed after SA treatment at different times. The other ten genes containing the TCA element were upregulated almost 2 h after SA treatment. These results suggest that SA might not only directly regulate the TCA element but also link other pathways to indirectly regulate the expression of some *TCP* genes.

Most *BjTCP* genes have the polytype GA response elements GARE, P-box, and TATC-box in their promoter regions, which might lead to more complex regulation of their expression and more diverse expression patterns. Analysis of the GA response elements of these genes showed no GA-related elements in the promoter regions of *BjTCP1a*, *BjTCP7a*, *BjTCP8a*, *BjTCP12a*, *BjTCP14a*, *BjTCP20a*, and *BjTCP22a*. After GA treatment, no expression of *BjTCP12a* and *BjTCP20a* was detected, while *BjTCP1a*, *BjTCP7a*, *BjTCP8a*, *BjTCP14a*, and *BjTCP22a* expression first increased and then decreased. In addition, although there was a P-box element in *BjTCP17a*, the expression level did not change after GA treatment. The remaining genes including one or more GA response elements in their promoter regions were upregulated after GA treatment. These results suggest that GA affects the expression levels of most of the *TCP* genes.

This study was the first to identify 62 *BjTCP* genes in *B. juncea* var. tumida and to investigate their roles in stem development. On the basis of our results and reports on *A. thaliana*, we believe that *BjTCP* is also regulated by many factors and is involved in hormone response, plant architecture, inflorescence development, and immune regulation in *B. juncea* var. tumida (Fig. 8). Our results will provide the foundation for further determining the molecular mechanism underlying stem swelling and flowering orchestrated by *TCP* genes in *B. juncea* var. tumida.

## CONCLUSIONS

We performed a genome-wide analysis and identified 62 *BjTCP* genes in *B. juncea* var. tumida. These genes are divided into two 34 class I and 28 class II subfamilies. Of these 62 *BjTCP* genes, 61 are heterogeneously distributed on 18 chromosomes, 51 have no introns, and most of the *BjTCP* genes in the same cluster have similar patterns of exon length, intron number, and conserved motifs. Several genes are highly expressed in the development of *B. juncea* var. tumida, and branching-related genes have low expression in the swelling stage of vegetative growth.

## ACKNOWLEDGEMENTS

We acknowledge Dr Yinghong Li for his assistance in RNA-seq data processing and related bioinformatics analysis.

### Funding

This work was supported by the Science and Technology Research Program of Chongqing Municipal Education Commission (Grant No. KJQN201800609) and Science and Technology Research Program of Chongqing University of Posts and Telecommunications (Grant No. A2018-114) and Chongqing Natural Science Foundation (Grant No. cstc2015jcyjA0752). The funders had no role in study design, data collection and analysis, decision to publish, or preparation of the manuscript.

### Grant Disclosures

The following grant information was disclosed by the authors:
Chongqing Municipal Education Commission: KJQN201800609.
Chongqing University of Posts and Telecommunications: A2018-114.
Chongqing Natural Science Foundation: cstc2015jcyjA0752.

### Competing Interests

The authors declare there are no competing interests.

### Author Contributions

- Jing He performed the experiments, prepared figures and/or tables, and approved the final draft.
- Xiaohong He and Huaizhong Jiang performed the experiments, authored or reviewed drafts of the paper, and approved the final draft.
- Pingan Chang and Daping Gong analyzed the data, authored or reviewed drafts of the paper, and approved the final draft.
- Quan Sun conceived and designed the experiments, analyzed the data, prepared figures and/or tables, and approved the final draft.

## Data Availability

Data is available at NCBI SRA: SRX108496, SRX108498–SRX108502.

## Supplemental Information

Supplemental information for this article can be found online at http://dx.doi.org/10.7717/peerj.9130#supplemental-information.

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
