# Peer review of "Genome-wide identification and characterization of TCP family genes in Brassica juncea var. tumida"

_PeerJ, doi:10.7717/peerj.9130_

## Round 0.1 · original submission · Major Revisions

Please carefully consider the comments from the reviewers. In articular, please note the comment that the phylogenetic analysis must be performed again using correct methods, as suggested and then update the results and discussion accordingly

Reviewer 1 ·

Basic reporting

The English language is clumsy and should be improved.
The literature is poorly explored. Several references are lacking, along with a better Discussion.
Figures and tables are in general ok (although I detail some concerns for the authors)

Experimental design

Ther study is relevant for the Journal scope. For this reason, a second chance should be given to the authors.
I have some methodological concerns that can be addressed by the authors.

Validity of the findings

I suggest that the findings should be further explored and analysed, specially miR319 targets within the TCP family.

Additional comments

The present study by He et al., reports a genome-wide characterization of TCP transcription factors in Brassica juncea var. Tumida, combining phylogenetics and gene expression data.
The approach of the manuscript may provide new valuable insights in line with the scope of the journal, and also may set the basis for future investigation of the function of TCP members in B. juncea. Nevertheless, I consider that the manuscript can not be accepted in its current version, but a new improved manuscript can be considered for publication in PeerJ.
My main concern is the phylogenetic analysis performed. The authors performed a neighbor-joining tree, which is a distance method, not a cladistic analysis. This is a common error within the comunity of plant molecular biologist. I strongly suggest that you perform a phylogenetic analysis through Maximum likelihood, Bayesian, or even Parsimony. This can change the topology of the tree presented in Fig 2. In fact, authors can use the ML o Parsimony search implemented in MEGA, which was the software of your choice. Lack of experience in phylogenetics is reflected by the fact that (according to the legend of Fig 2A) you presented the boostrap tree, which only serve to obtain clade support, but not to be used as the final topology of a phylogenetic analysis. This is a conceptual mistake. Given that the gene cluster obtained may change after you perform a phylogenetic analysis, the outcome and conclusions may also suffer modifications.
There also other relevant points that need consideration:
1- Literature used is not extensive as it should be, taking into account the knowledge about TCP nowadays.
2- The English language is clumsy and should be highly improved to ensure that an international audience can clearly understand your ms. Although I am not an English native speaker person, some parts of the text were extremely difficult to reach your point. I made the effort of suggest some corrections (see minor comments & corrections), but I consider this is a major pitfall of the ms.
3- Authors claim that in B. juncea there are no homologous genes of AtTCP3, 4, 10, which are direct targets of miR319. These should be further explored given the possibility that TCP family in this species seems to scape the post-transcriptional control by miR319. Authors should confirm this result and discuss it further. In case you find some BjTCP target candidates for miR319, you should at least present an alignement in a figure.
Minor comments & corrections:
Title: replace “characters” with “characterization”.
Introduction:
P6, line 35
…that is reported to BE involved
Also, wrong use of the singular
P6, line 41
Missing space. The same mistake is everywhere in the text.
loop(
P6, line 44
The core sequences described are essentially the same (GGNCCC). Also, class I site is incorrect according to recent reports of TCP, revise with updated literature.
P6, line 48
Misspelling; controlingl
P6, line 51
Incorrect interpretation of Niwa et al. 2013. Although they proved FT and BRC1 interaction, they do not evaluate the importance of this interaction in floral development. Rephrase
P6, line 52
binding with. Incorrect preposition
P7, line 65
Remove the s after sativa
P7, line 87
Awkward use of English.
Suggestion for improvement:
SINCE As the whole genome of tumorous stem mustard was sequenced (Yang et al. 2016), a genome wide analysis of TCP genes was PERFORMED FOR the first time to perform in this research.
P8, line 92
The results can provide valuable information for the classification of BjTCPs and lay the foundation FOR of exploring the mechanism of stem swelling and flowering TIMEregulation regulated ORCHESTATED by TCP proteins in tumorous stem mustard.
Material and Methods:
P8 line 98.
What do you mean with constant light? A 16/8 h light/dark photoperiod is not constant light!. You may refer to temperature instead.
P8 line 103.
No N2 snap frozen first?
P8 line 112.
Use plural in sequenceS
Use past tense for donwload
P8 line 123.
…a maximum number of 12 motifs WAS SET.
P8 line124
Reference for ProComp9.0??
P8 line126
Use cursive for “cis-acting”
P8 line131
Reference for MapInspect?
P8 line133
…WAS can downloaded…
P8 line134
accession number were SRX108496
Results
P10 line 167
…were ARE transcription.
The generalized conclusion of line 166-168 must be rephrased, considering that the authors found some BjTCPs outside the nuclei. In addition, what do you mean in Table 1 with “none” in subcellular localization for BjTCP13a-c?. You didn´t found the protein inside the cell at all? This point should be further explained. I am not familiarized with ProComp9.0, but it is expected that the protein is present somewhere in the cell. This data comes from empirical evidence or is just an in silico prediction?.
P11 Line 179
The absence of homologous sequences of AtTCP2-4 and 10 may be interesting to explore further considering that these proteins are targets of miR319. Other genome wide analyses reported of TCP proteins usually check the conservation of miR319 targets (even in plant species outside the Brassicaceae family). I strongly suggest you do the same. In case you confirm that there are no miR319 targets in BjTCP, these would become an interesting outcome (that should be explored and disscused). Also, under this scenario, it may be relevant also to check the existence of miR319 in Bj.
P11 Line 183-185
The grammar in this phrase is clumsy. Also, you must avoid the use of constrictions like “couldn’t”.
P12 line 208
…and BjTCP18a-d were HAVE more…
P12 line 209
“Chinses”???
P12 line 209
Use italics for BrTCP1a,b (and everywhere in the text)
P12 line 209
exon number form FROM other
P12 line 208-211
The grammar of this phrase should be revised.
P12 line 212
their homologous genes IN B. rapa BrTCP18a and b
Figure 3: Indicate which motif/s include the TCP domain. Remove the Green box with the legend CDS in 3C.
Also, did BjTCP21b have the TCP domain???
P12 line 231
was WERE also present
P13 line 237
were also be found
line 242
on A the previous
line 245
“DA simple” means DY samples?
Line 255
The criterion used to select genes for qPCR is correct, if the sequence similarity is at the promoter regions particularly. This should be explicit.
Line 258-259
Missing Word before “expected”?? AS??
DISCUSSION
Line 273
..some genes were duplicated are doubled in the..
Line 277 ..identified. We…
Line 282-284
It is difficualt to interpret what is the author’s point, given grammar errors.
Line 287 (suggestion for improvement)
…we guess SPECULATE THAT these similarities WITHIN cluster OF homologous genes members SUGGEST THAT THEY might play similar function during tumorous stem mustard growth and development.
Line 291
No need to cite Figures in Discussion section, unless this is indicated in the instruction for authors of the Journal.
Line 294
Given the high copy number through duplication of TCP15 in Bj, along with the expression data analysed, it would be interesting to discuss with the reported functions of AtTCP15.
Suggestion for improvement
AS MENTIONED ABOVE, Tthere are 24 TCP genes in Arabidopsis. Some corresponding homologous OF TCP genes WERE cannot fOUnd in tumorous stem mustard, such as TCP2-4, 6, 11 and 16, PROBABLY DUE TO GENE LOSTwhich may be caused by the losing eventS during the evolutionARY process.
Lines 299-303
This claim is incorrect: “In theory, each of the Arabidopsis TCP genes has two homologous genes in the stem mustard...”
Why Bj has to have the double of genes than Arabidopsis?? Take into account thay the lineage of the family where both species belongs split long ago. Remove or rephrase!
Line 303
“doubling”? replace with duplication.
Also, a connector is missing after “,”
Line 313
were demonstrated REPORTED for
Suggestion for improvement: BRC1/TB1 orthologS with TB1 and rice TB1 that may play a similar role in development of the primary shoot architecture
Line 320
But There are four BjTCP18s homologous genes in tumorous stem mustard, which may HAVE BEEN formED by gene duplication.
Functional differentiation may occur between the four TCP18 genes. From GIVEN their DIFFERENTIAL expression patterns, these four genes are not consistent during tissue development.
LINE328
INDICATEd
Line 329
Suggestion for improvement (if I get your idea)
According to reports, there are 16 varieties of mustard species identified AND USED FOR FOOD CONPSUMPTION, in which the mainly difference of these vegetables tissues are used for food and shape, including root, stem, leaf and..
Also, have you found a report were says that BRC1 regulates flowering time?
Line 335
expressionED tan
Line 338
A lot of references are missing. Authors should explore more deeply the literature about TCPs and hormones.
Line 350, suggestion:
The OTHER TEN GENES rest containing the TCA element ten genes are upregulated
Line 351
directly regulated on the TCA element, but also causesMAY LINK other pathways to INDIRECTLY coordinately regulate the expression of the SOME TCP geneS.
Line 358
was not detectED

Annotated reviews are not available for download in order to protect the identity of reviewers who chose to remain anonymous.

Reviewer 2 ·

Basic reporting

MS needs to be screened for typo errors.

Experimental design

Experimental design is good. Some information needs to be mentioned in methodology. See my detailed comments in "comments to author" section

Validity of the findings

Fine but authors are suggested to make a putative mechanism diagram based on current investigation.

Additional comments

Manuscript by Jing He et al. is an interesting one, but before its further consideration, authors should amend below mentioned suggestions:
Comments/Suggestions
Line 19, use full botanical name at first appearance.
Line 19, “thirty-three” within a sentence use numerical values to denote numbers. Be uniform throughout the MS.
Line 26,mention full forms of abbreviations used at first appearance . for example “YA strains. After treatment with GA and SA” Be uniform throughout MS.
Line 34-38, provide reference for specific parameter. For example : embryonic growth (reference), leaf
Development (reference), branching (reference), flowering (reference), circadian rhythm (reference), hormone signalling (reference),
Materials and Methods
Mention all growth conditions in detail.
What is the rationale behind selection of particular concentrations for hormones? Mention it.
What were qRT-PCR conditions?
Mention original reference for “The relative gene expression level was calculated using the 2-ΔΔCt method.”

Authors are suggested to make a putative mechanism diagram on the basis of current results and discussion. Cite that figure in conclusion section. This figure will be very informative for readership.

---

## Round 0.2 · Major Revisions

Many thanks for addressing the reviewers concerns and improving the manuscript. However In this day and age of high-volume gene analysis there is a demand to add metadata to help researchers make use of the data presented. As this work is classifying 62 genes and characterizing the expression under different tissue and condition constraints, the members need to be evaluated into ontological terms. Especially since there are putative mechanisms proposed; information would be readily available to continue with the data presented.

Journal manuscripts are often scanned by text-mining software that locates and extracts core data elements, like gene function. Adding standard ontology terms, such as the Gene Ontology (GO, geneontology.org) or others from the OBO foundry (obofoundry.org) can enhance the recognition of your contribution and description. This will also make human curation of literature easier and more accurate. None of this was visible.

There were no specifics on where the sequence data is derived except for the start (not star) and stop locations listed in Table 1. The BRAD site does have some Brassica juncea sequences, but the manuscript does not stipulate either the V1.1 or V1.5 versions, and the site does not define var. tumida. Additional clarity is required.

In summary, the manuscript requires better preparation.

Reviewer 1 ·

Basic reporting

The authors provide an improved revised manuscript. Overall, the revisions have clarified issues that now enabled a more informed evaluation of the manuscript. The authors have responded to my queries. They have largely addressed them convincingly yet important concerns.

Experimental design

The experimental design is ok now.

Validity of the findings

Authors now provides a better disccusion of their findings.

Reviewer 2 ·

Basic reporting

No comments

Experimental design

No comments

Validity of the findings

No comments

Additional comments

N/A

---

## Round 0.3 · accepted · Accept

Many thanks for making the required changes.